# Adenosine A_1_-A_2A_ Receptor-Receptor Interaction: Contribution to Guanosine-Mediated Effects

**DOI:** 10.3390/cells8121630

**Published:** 2019-12-13

**Authors:** Débora Lanznaster, Caio M. Massari, Vendula Marková, Tereza Šimková, Romain Duroux, Kenneth A. Jacobson, Víctor Fernández-Dueñas, Carla I. Tasca, Francisco Ciruela

**Affiliations:** 1Programa de Pós-graduação em Neurociências, Centro de Ciências Biológicas, Universidade Federal de Santa Catarina, 88040-900 Florianópolis, Brazil; de_lanz@hotmail.com; 2Programa de Pós-graduação em Bioquímica, Centro de Ciências Biológicas, Universidade Federal de Santa Catarina, 88040-900 Florianópolis, Brazil; caio.massari@gmail.com; 3Unitat de Farmacologia, Departament de Patologia i Terapèutica Experimental, Facultat de Medicina i Ciències de la Salut, IDIBELL, Universitat de Barcelona, 08907 L’Hospitalet de Llobregat, Spain; Vendy.Markova@seznam.cz (V.M.); simkovat@gmail.com (T.Š.); 4Institut de Neurociències, Universitat de Barcelona, 08035 Barcelona, Spain; 5Molecular Recognition Section, Laboratory of Bioorganic Chemistry, National Institute of Diabetes and Digestive and Kidney Diseases, National Institutes of Health, Bethesda, MD 20892, USA; Romain.Duroux@niddk.nih.gov (R.D.); kennethj@niddk.nih.gov (K.A.J.); 6Departamento de Bioquímica, Centro de Ciências Biológicas, Universidade Federal de Santa Catarina, 88040-900 Florianópolis, Brazil

**Keywords:** guanosine, neuroprotection, oxygen/glucose deprivation, NanoBRET, A_1_R/A_2A_R heteromer

## Abstract

Guanosine, a guanine-based purine nucleoside, has been described as a neuromodulator that exerts neuroprotective effects in animal and cellular ischemia models. However, guanosine’s exact mechanism of action and molecular targets have not yet been identified. Here, we aimed to elucidate a role of adenosine receptors (ARs) in mediating guanosine effects. We investigated the neuroprotective effects of guanosine in hippocampal slices from A_2A_R-deficient mice (A_2A_R^−/−^) subjected to oxygen/glucose deprivation (OGD). Next, we assessed guanosine binding at ARs taking advantage of a fluorescent-selective A_2A_R antagonist (MRS7396) which could engage in a bioluminescence resonance energy transfer (BRET) process with NanoLuc-tagged A_2A_R. Next, we evaluated functional AR activation by determining cAMP and calcium accumulation. Finally, we assessed the impact of A_1_R and A_2A_R co-expression in guanosine-mediated impedance responses in living cells. Guanosine prevented the reduction of cellular viability and increased reactive oxygen species generation induced by OGD in hippocampal slices from wild-type, but not from A_2A_R^−/−^ mice. Notably, while guanosine was not able to modify MRS7396 binding to A_2A_R-expressing cells, a partial blockade was observed in cells co-expressing A_1_R and A_2A_R. The relevance of the A_1_R and A_2A_R interaction in guanosine effects was further substantiated by means of functional assays (i.e., cAMP and calcium determinations), since guanosine only blocked A_2A_R agonist-mediated effects in doubly expressing A_1_R and A_2A_R cells. Interestingly, while guanosine did not affect A_1_R/A_2A_R heteromer formation, it reduced A_2A_R agonist-mediated cell impedance responses. Our results indicate that guanosine-induced effects may require both A_1_R and A_2A_R co-expression, thus identifying a molecular substrate that may allow fine tuning of guanosine-mediated responses.

## 1. Introduction

Guanosine is a guanine-based purine nucleoside that has been shown to exert neuroprotective and neurotrophic effects in both in vitro and in vivo studies (for review, see [1]). Thus, it has been postulated as a good candidate for the management of several central nervous system (CNS) disorders, including neurodegenerative diseases (i.e., Parkinson’s, Alzheimer’s) or ischemia [1,2]. Brain ischemia is one of the major health disability conditions worldwide [3]. It occurs after a blood supply collapse that leads to a reduced level of oxygen and glucose within the affected brain area. Similarly, upon excitotoxicity and oxidative stress a failure of cellular bioenergetics occurs [4]. Importantly, a neuroprotective role of guanosine has been extensively investigated in animal and cellular models of ischemia, excitotoxicity and oxidative stress [5,6,7,8,9,10]. Indeed, we have demonstrated that guanosine prevents reactive oxygen species (ROS) generation and cell death in hippocampal slices subjected to the oxygen/glucose deprivation (OGD) [11].

The mechanism by which guanosine exerts its neuroprotective effects is still intriguing. Despite the identification of a putative guanosine binding site in rat brain membranes [12], a specific guanosine receptor has not yet been discovered. Importantly, it has been hypothesized that adenosine receptors (ARs) may play a role in mediating guanosine effects, although with some controversy. For instance, it has been reported that AR selective ligands do not compete for guanosine binding to rat brain membranes [13,14], whereas AR ligands were able to block some of the guanosine-dependent neuroprotective effects [15]. In line with this, a selective adenosine A_1_ receptor (A_1_R) antagonist (DPCPX, 8-cyclopentyl-1,3-dipropylxanthine) and a selective A_2A_ receptor (A_2A_R) agonist (CGS21680, 2-(4-(2-carboxyethyl)phenethylamino)-5′-*N*-ethylcarboxamidoadenosine) inhibited guanosine-mediated neuroprotection in hippocampal slices subjected to OGD [11]. Overall, these findings, including those using multimodal A_1_R and A_2A_R ligand treatments, supported the notion that both A_1_R and A_2A_R would participate in guanosine-mediated effects. 

Interestingly, it has been hypothesized that adenosine A_1_ and A_2A_receptor-receptor interactions (i.e., heteromerization) might be behind some of the guanosine-mediated effects, thus pointing to the A_1_R/A_2A_R heteromer as a putative molecular target for guanosine [16]. Indeed, the existence of A_1_R/A_2A_R heteromers has been demonstrated in presynaptic terminals of striatal neurons controlling glutamate release [17], thus acting as an adenosine concentration-dependent switch [18]. Consequently, low to moderate concentrations of adenosine predominantly activate A_1_R within the A_1_R/A_2A_R heteromer (i.e., inhibiting glutamate release), whereas moderate to high concentrations of adenosine also activate A_2A_R, which, by means of the A_1_R-A_2A_R intramembrane negative allosteric interaction, antagonizes A_1_R function, therefore facilitating glutamate release. Altogether, in view of the already known experimental indications, the A_1_R/A_2A_R heteromer might be viewed as a potential target for guanosine, thus deserving further attention. Here, we aimed to assess the role of A_1_R and A_2A_R interaction in guanosine-mediated effects. First, we studied the neuroprotective effects of guanosine in an ex vivo model of brain ischemia, both in wild-type and A_2A_R deficient (A_2A_R^−/−^) mice; subsequently, we aimed to elucidate, in vitro, both the putative guanosine binding and activation of the A_1_R/A_2A_R heteromer.

## 2. Materials and Methods

### 2.1. Chemicals

The ligands used were: adenosine and guanosine from Sigma-Aldrich (St. Louis, MO, USA); CGS21680 and SCH442416 (2-(2-furyl)-7-[3-(4-methoxyphenyl)propyl]-7*H*-pyrazolo [4,3-*e*]-[1,2,4]triazolo [1,5-*c*]pyrimidin-5-amine) from Tocris Bioscience (Ellisville, MI, USA). Adenosine deaminase (ADA) was purchased from Roche Diagnostics (GmbH, Mannheim, Germany) and zardaverine from Calbiochem (San Diego, CA, USA). MRS7396, which is a selective fluorescent antagonist at the A_2A_R derived from SCH442416, was previously described [19].

### 2.2. Animals

Wild-type and A_2A_R^−/−^ CD-1 male and female mice [20] weighing 25–50 g were used at 2–3 months of age. The University of Barcelona Committee on Animal Use and Care (CEEA-UB) approved the protocol (Code 10033, 04/02/2018). Animals were housed and tested in compliance with the guidelines described in the Guide for the Care and Use of Laboratory Animals [21] and following the European Union directives (2010/63/EU), FELASA and ARRIVE guidelines. Mice were housed in groups of five in standard cages with ad libitum access to food and water and maintained under a 12-h dark/light cycle (starting at 7:30 AM), 22 °C temperature, and 66% humidity (standard conditions).

### 2.3. OGD Protocol

Mice were euthanized by cervical dislocation and hippocampi rapidly removed and placed in an ice-cold Krebs-Ringer bicarbonate buffer (KRB) (composition in mM: 122 NaCl, 3 KCl, 1.2 MgSO_4_, 1.3 CaCl_2_, 0.4 KH_2_PO_4_, 25 NaHCO_3_ and 10 d-glucose). The buffer was bubbled with 95% O_2_/5% CO_2_ up to pH 7.4. Slices (0.3 mm) were prepared using a Leica VT1200 vibrating blade microtome (Leica, Wetzlar, Germany) in KRB at 4 °C, and one slice per tube was allowed to recover for 30 min in KRB at 37 °C. Control hippocampal slices were incubated until the end of the experiment (15 min plus 2 h) in oxygenated KRB. OGD was induced by incubating the slices for a 15 min period in an OGD buffer in Hank’s balanced salt solution (HBSS; composition in mM: 1.3 CaCl_2_, 137 NaCl, 5 KCl, 0.65 MgSO_4_, 0.3 Na_2_HPO_4_, 1.1 KH_2_PO_4_, and 5 HEPES), where 10 mM d-glucose was replaced by 10 mM 2-deoxy-glucose and equilibrated with a 95% N_2_/5% CO_2_ gas mixture, as described previously [5] After 15 min of OGD the media of the slices was replaced by oxygenated KRB and maintained for 2 h for evaluation of cellular viability and ROS generation. Guanosine (100 µM), when present, was added 15 min before (in KRB) and during OGD (in OGD buffer), and maintained in the re-oxygenation period (2 h), when the OGD buffer was replaced by physiological KRB.

### 2.4. Cellular Viability Evaluation

For cellular viability assessment, slices were incubated in 0.5 mg/mL 3-(4,5-dimethylthiazol-2-yl)-2,5-diphenyltetrazolium bromide (MTT) (Sigma-Aldrich) for 20 min at 37 °C, as previously described [22]. In brief, the tetrazolium ring of MTT is first cleaved by active dehydrogenases to produce a precipitated formazan. Then, precipitated formazan can be solubilized with 200 μL of dimethyl sulfoxide (DMSO) and cellular viability quantified spectrophotometrically at a wavelength of 550 nm by means of a POLARstarplate-reader (BMG Labtech, Durham, NC, USA).

### 2.5. Measurement of ROS Production

For evaluating ROS generation, slices were incubated with 80 µM 2′,7′-dichlorofluorescein diacetate (DCFH-DA; Sigma-Aldrich) for 30 min [23]. Then, subsequent to the OGD/reoxygenation protocol, slices were washed twice with KRB and maintained for 15 min before adding DCFH-DA. H_2_DCFDA diffuses through the cell membrane, and it is hydrolyzed by intracellular esterases to the non-fluorescent form dichlorofluorescin (DCFH). Afterwards, DCFH can react with intracellular H_2_O_2_ to form dichlorofluorescein (DCF), a green fluorescent dye. Slices were then transferred to a 96-well black plate containing 200 µL of KRB, and fluorescence was read (excitation 480 nm, emission 525 nm) using a POLARStar plate reader (BMG Labtech).

### 2.6. Plasmid Constructs

The cDNA encoding the human A_1_R tagged at its N-terminal tail with the O6-alkylguanine-DNA alkyltransferase (i.e., A_1_R^SNAP^) cloned in pRK5 vector (BD PharMingen, San Jose, CA, USA) was a gift from Prof. Jean-Philippe Pin (CNRS, Montpellier, France). Thus, to perform functional assays A_2A_R^SNAP^ [24] and A_1_R^SNAP^ were used. Also, A_2A_R^RLuc^ and A_1_R^YFP^ constructs [17] were used to perform classical BRET (Bioluminescence Resonance Energy Transfer) assays. Finally, to perform NanoBRET experiments with the MRS7396 fluorescent antagonist, we created an A_2A_R NanoLuc sensor (A_2A_R^NL^). To this end, the cDNA encoding the human A_2A_R was amplified by polymerase chain reaction from the pECFP-A_2A_R vector using the primers: FA2AEco (5′-GCCG**GAATTC**CCCATCATGGGCTCCTCGGTGTAC-3′) and RA2ANot (5′-CGCG**GCGGCCGC**tcaggacactcctgctccatcctggg-3′). The amplified A_2A_R insert was then cloned into the *Eco*RI/*Not*I sites of pNLF1-secN vector (Promega, Stockholm, Sweden) containing a hemagglutinin (HA) epitope tag. All the constructs were verified by DNA sequencing.

### 2.7. Cell Culture and Transfection

Human embryonic kidney (HEK)-293T cells were grown in Dulbecco’s modified Eagle’s medium (DMEM) (Sigma-Aldrich), supplemented with 1 mM sodium pyruvate, 2 mM L-glutamine, 100 U/mL streptomycin, 100 mg/mL penicillin and 5% (*v*/*v*) fetal bovine serum at 37 °C and in an atmosphere of 5% CO_2_. HEK-293T cells growing in 60 cm^2^ plates were transfected with the cDNA encoding the different plasmids using linear PolyEthylenImine reagent (PEI) (Polysciences Inc., USA).

### 2.8. NanoBRET Experiments

The NanoBRET assay was performed on stably expressing (A_2A_R^NL^) HEK-293T cells, transiently transfected (or not) with A_1_R^SNAP^, according to [25]. In brief, cells were re-suspended in HBSS, and seeded into poly ornithine coated white 96-well plates. After 24 h, cells were challenged with/without the non-labelled A_2A_R antagonist (SCH442416) or guanosine and incubated for 1 h at 37 °C. Subsequently, the fluorescent ligand (MRS7396) was added and the plate and returned to 37 °C for 1 h. Finally, coelenterazine-h (Life Technologies Corp.) was added at a final concentration of 5 μM, and readings were performed after 5 min using a CLARIOStar plate reader (BMG Labtech). The donor and acceptor emissions were measured at 490–510 nm and 650–680 nm, respectively. The raw NanoBRET ratio was calculated by dividing the 650 nm emission by the 490 nm emission. In competition studies, results were expressed as a percentage of the maximum signal obtained (mBU; milliBRET Units).

### 2.9. cAMP Assay

cAMP accumulation was measured using the LANCE^®^ Ultra cAMP Kit (PerkinElmer, Waltham, MA, USA) as previously described [26]. In brief, transfected (A_2A_R^SNAP^ or A_2A_R^SNAP^ + A_1_R^SNAP^) HEK-293T cells were firstly incubated for 1 h at 37 °C with stimulation buffer (BSA 0.1%, ADA 0.5 units/mL, zardaverine 2 µM; in serum-free DMEM) and later on with CGS21680 for 30 min at 37 °C. Thereafter, cells were transferred to a 384-well plate in which reagents were added following manufacturer’s instructions. After 1 h at room temperature, Time-Resolved-Fluorescence Resonance Energy Transfer (TR-FRET) was determined by measuring light emission at 620 nm and 665 nm by means of a CLARIOstar plate reader (BMG Labtech).

### 2.10. Intracellular Calcium Determinations

The A_1_R-mediated intracellular Ca^2+^ accumulation was assessed by means of a luciferase reporter assay based on the expression of the nuclear factor of activated T-cells (NFAT), as previously described [27]. In brief, cells were transfected with the cDNA encoding the A_1_R, the NFAT-luciferase reporter (pGL4-NFAT-RE/luc2p; Promega) and the yellow fluorescent protein (pEYFP-N1; Promega). After 36 h post-transfection, cells were incubated with the indicated drugs for 6 h. Subsequently, cells were harvested with passive lysis buffer (Promega), and the luciferase activity of cell extracts was determined using a luciferase Bright-Glo^TM^assay (Promega) in a POLARStar plate-reader (BMG Labtech) using a 30-nm bandwidth excitation filter at 535 nm.

### 2.11. Label-Free Cellular Impedance Assay

The xCELLigence Real-Time Cell Analyzer (RTCA) system (ACEA Biosciences, San Diego, CA, USA) was employed to measure changes in cellular impedance correlating with cell spreading and tightness, thus being widely accepted as a morphological and functional biosensor of cell status [28,29,30]. Thus, 16-well E-plates (ACEA Biosciences) were coated with 50 μL fibronectin (10 μg/mL) at 37 °C for 1 h before being washed three times with 100 μL MilliQ-water before use. The background index for each well was determined with 90 μL of stimulation buffer (supplemented DMEM with ADA 0.5 U/mL and zardaverine 10 μM) in the absence of cells. Data from each well were normalized to the time point just before compound addition using the RTCA software providing the normalized cell index (NCI). Subsequently, HEK-293T cells permanently expressing the A_2A_R^SNAP^ construct [31] in the absence or presence of A_1_R^SNAP^ (90 μL resuspended in stimulation buffer) were then plated at a cell density of 40,000 cells/well and grown for 18 h in the RTCA SP device station (ACEA Biosciences) at 37 °C and in an atmosphere of 5% CO_2_ before ligand (i.e., CGS21680 and/or guanosine) addition. Cell index values were obtained immediately following ligand stimulation every 15 s for a total time of at least 50 min. For data analysis, the area under the curve (AUC) for each NCI trace response was quantified and normalized to the basal. 

### 2.12. Statistics

Data are represented as mean ± standard error of mean (SEM). The number of samples/animals (*n*) in each experimental condition is indicated in the corresponding figure legend. Comparisons among experimental groups were performed by Student’s *t-*test and ANOVA, using GraphPad Prism 6.01 (San Diego, CA, USA), as indicated. Statistical difference was accepted when *p* < 0.05. 

## 3. Results

### 3.1. Guanosine-Mediated Neuroprotection in Hippocampal Slices Depends on A_2A_R Expression

It has been postulated that ARs might be involved in guanosine-mediated responses in vivo [16]. Within this line of inquiry, we first interrogated whether A_2A_R expression is necessary for guanosine-mediated neuroprotection, a well-known guanosine effect in vivo [1]. To this end, we subjected hippocampal slices from wild-type (i.e., A_2A_R^+/+^) and A_2A_R^−/−^ mice to an OGD protocol in the presence or absence of guanosine. Indeed, significant cell death (*p* < 0.001) and ROS production (*p* = 0.0359) were observed in A_2A_R^+/+^ hippocampal slices subjected to the OGD protocol (Figure 1A,B). Interestingly, guanosine (100 µM) was able to prevent these effects, thus cellular viability significantly increased (*p* = 0.0012) and ROS production decreased (*p* = 0.0389) (Figure 1A,B), as previously reported [5,11]. Importantly, under the same experimental conditions, in hippocampal slices obtained from A_2A_R^−/−^ mice, guanosine failed to prevent OGD-mediated cell death (*p* = 0.005) and ROS production (*p* = 0.0279) (Figure 1A,B), thus losing its neuroprotective effect. Overall, these results suggested that A_2A_R expression was necessary for guanosine-mediated neuroprotection.

### 3.2. A_2A_R Ligand Binding is Affected by Guanosine upon A_1_R Coexpression

Once we demonstrated that the neuroprotective effect of guanosine was A_2A_R-dependent, we aimed to assess the putative direct interaction of guanosine with A_2A_R through ligand binding studies. To this end, we engineered a fluorescent ligand BRET-based assay to assess A_2A_R ligand binding in living cells (Figure 2A). We used a fluorescent A_2A_R antagonist (MRS7396) that is able to engage in a BRET process upon interacting with a cell surface A_2A_R tagged with the NanoLuciferase (NL) at its N-terminus (i.e., A_2A_R^NL^) (Figure 2A). MRS7396 is a BODIPY630/650 derivative of SCH442416 [19], which upon A_2A_R binding can act as an acceptor chromophore for NanoLuciferase emission (490 nm) in a BRET process. Thus, we challenged stable A_2A_R^NL^-expressing cells with increasing concentrations of MRS7396, in the presence/absence of non-labelled SCH442416. Interestingly, a bell-shaped binding saturation hyperbola, with a K_D_ = 4.8 ± 2.7 nM, was obtained for MRS7396, while in the presence of a saturating concentration of SCH442416 (1 µM) the binding was displaced (Figure 2B). Our results showed that the NanoBRET binding assay was a robust and reliable way to assess A_2A_R ligand binding. Accordingly, we next assessed possible guanosine effects on A_2A_R orthosteric binding by performing a competition assay with a fixed concentration of MRS7396 (10 nM) (occupying ~80% of receptors at equilibrium) and increasing concentrations of guanosine. Interestingly, under these experimental conditions, guanosine was unable to alter MRS7396 binding to A_2A_R^NL^ (Figure 2C), thus indicating that guanosine does not orthosterically bind to A_2A_R, as previously reported [12,13].

Since A_2A_R heteromerizes with A_1_R [17], and some of the physiological effects of guanosine were modulated by A_1_R ligands [32,33], we investigated whether A_1_R/A_2A_R heteromer formation affected AR-related guanosine-dependent effects. To this end, we first recreated the formation of A_1_R/A_2A_R heteromers in HEK-293T cells by transfecting A_2A_R^RLuc^ and A_1_R^YFP^ constructs and monitoring A_2A_R/A_1_R heteromerization by a classical BRET approach (Figure A1). Interestingly, neither adenosine nor guanosine incubation altered A_1_R/A_2A_R heteromer formation (Figure A1). Subsequently, we assessed the impact of A_1_R co-expression in A_2A_R binding of MRS7396 using our NanoBRET binding assay. Notably, in A_1_R-A_2A_R doubly expressing cells, guanosine (100 µM) was able to significantly reduce by 19 ± 4% (*p* = 0.0138) the binding of MRS7396 to the A_2A_R^NL^, thus indicating that the A_1_R/A_2A_R heteromer might play a potential role in AR-related guanosine-dependent effects (Figure 2C).

### 3.3. A_2A_R Signalling, but Not A_1_R, is Modulated by Guanosine in an A_1_R Coexpression-Dependent Manner

Given that guanosine reduced A_2A_R binding in an A_1_R-expression-dependent manner, we next aimed to determine whether guanosine also impinged into A_2A_R signaling. Accordingly, we determined the effects of guanosine in A_2A_R-mediated cAMP accumulation upon agonist incubation. In A_2A_R-expressing cells, the selective A_2A_R full agonist CGS21680 induced a concentration-dependent cAMP accumulation (pEC_50_ = 7.98 ± 0.08), indicating that the receptor was expressed and functional at the plasma membrane (Figure 3A). Subsequently, we challenged cells with a fixed concentration of CGS21680 (200 nM) and evaluated the effects of increasing concentrations of guanosine in A_2A_R-dependent cAMP accumulation. As shown in Figure 3B, guanosine did not preclude A_2A_R-mediated cAMP accumulation. Conversely, in cells doubly expressing A_1_R and A_2A_R, guanosine (100 µM) was able to significantly reduce, by 19 ± 3% (*p* = 0.0460), the A_2A_R-mediated cAMP accumulation (Figure 3B). These results supported the hypothesis that the effects of guanosine might be dependent on an A_1_R-A_2A_R interaction.

Interestingly, our NanoBRET-based binding results and cAMP determinations in the absence and presence of A_1_R suggested a direct involvement of this receptor in guanosine-mediated blockade of A_2A_R ligand binding and signaling. Thus, to ascertain whether guanosine would directly interact with A_1_R we assessed its impact on A_1_R-dependent signaling. To this end, A_1_R-mediated calcium responses in HEK-293T cells were determined through a homogenous bioluminescence reporter assay system using a NFAT response element controlling luciferase gene expression. While the activation of A_1_R, via application of the agonist N^6^-*R*-phenylisopropyladenosine (*R*-PIA, 50 nM), increased intracellular Ca^2+^, the incubation with guanosine (100 μM) did not promote intracellular Ca^2+^ mobilization (Figure 4A). Similarly, when A_1_R-expressing cells were treated with R-PIA in the presence of increasing concentrations of guanosine, A_1_R-dependent intracellular Ca^2+^ mobilization was not affected, as observed in doubly A_1_R and A_2A_R transfected cells (Figure 4B). Overall, these results indicated that guanosine did not interact with A_1_R, thus ruling out any orthosteric A_1_R-dependent trans-inhibition of A_2A_R function in A_1_R-A_2A_R expressing cells.

Finally, we assessed the functional activity of guanosine using the label-free technology. To this end, the whole-cell guanosine-mediated impedance responses were monitored in living cells expressing A_2A_R in the absence or presence of A_1_R using a biosensor method, as previously reported [34]. First, we tested CGS21680-mediated changes in morphology (i.e., impedance) of A_2A_R^SNAP^ expressing HEK-293T cells, which were recorded in real-time. Interestingly, addition of CGS21680 resulted in a significant (*p* = 0.015) increase of impedance, which was blocked by incubation with the selective A_2A_R antagonist ZM241385 (Figure 5A,B). In addition, guanosine did not affect the cell basal morphology (*p* = 0.6105) nor its CGS218680-mediated changes (*p* = 0.1217) (Figure 5B). However, in doubly expressing A_1_R/A_2A_R cells guanosine significantly reduced (*p* < 0.0106) cell basal morphology and precluded (*p* < 0.0001) the CGS218680-induced increase in cellular impedance (Figure 5B). Again, these results indicated that the A_1_R-A_2A_R co-expression may play a potential role in AR-related guanosine-dependent cellular effects.

## 4. Discussion

Guanosine is a purine nucleoside with widely demonstrated extracellular neuromodulatory effects in the CNS, but so far without an identified receptor. Based on the use of selective ligands, ARs have been proposed as possible targets to explain guanosine-mediated effects in animal and cellular models of ischemia. However, at present, the mechanism of action of guanosine is not clear. Here, we show that A_2A_R expression was crucial for guanosine-mediated protective effects in an ex vivo model of brain ischemia. In addition, when examining guanosine effects in a controlled heterologous system, we were able to reveal the importance of a proposed A_1_R-A_2A_R interaction mediating guanosine effects, both in A_2A_R-ligand binding and in receptor function.

In the OGD ischemia model in hippocampal slices, we previously showed that guanosine induced a neuroprotective effect (increase of glutamate uptake) that was inhibited by activation of A_2A_R by CGS2180 [11]. This effect of CGS21680 in abolishing a guanosine-evoked increase in glutamate uptake in an OGD protocol was also observed in cultured astrocytes expressing the astrocytic glutamate transporter Glt-1 [15]. Therefore, here we evaluated guanosine’s neuroprotective effects in A_2A_R^−/−^ mice and revealed an important role for this receptor. Thus, in A_2A_R^−/−^ hippocampal slices, we observed a loss of the neuroprotective effects of guanosine (increasing viability and controlling ROS production in OGD conditions) that were observed in slices from wild-type mice (Figure 6A). This result, consistent with previous data, pointed to ARs as possible targets for guanosine [35,36], prompting us to further explore the mechanism by which guanosine might act.

Our NanoBRET-based sensor data suggested that, as previously reported [13], guanosine apparently does not bind directly to the A_2A_R. However, in A_1_R/A_2A_R cells, it was possible to observe a guanosine-mediated partial displacement of A_2A_R-ligand binding (Figure 6B). Together with the ex vivo data, this result would indicate that the mechanism of action of guanosine would be mediated by this receptor–receptor entity. Indeed, previous data showing both DPCPX- and pertussis toxin-dependent blockade of protective effects of guanosine in hippocampal slices subjected to OGD [11], supported the dependence on functional A_1_Rs coupled to a G-protein to mediate guanosine effects.

We found that guanosine reduced A_2A_R orthosteric binding only in A_1_R-A_2A_R expressing cells. Thus, we evaluated whether guanosine could modulate A_2A_R-dependent signaling under the same experimental conditions. Interestingly, while guanosine did not preclude CGS21680-induced cAMP accumulation in A_2A_R-expressing cells, it reduced A_2A_R-mediated cAMP accumulation in doubly A_1_R-A_2A_R transfected cells, as observed in the ligand-binding assay (Figure 6B). Additionally, the evaluation of guanosine effects on the functional activity of ARs using the label-free technology confirmed that guanosine-mediated cell impedance responses were dependent on A_1_R-A_2A_R co-expression. Hence, our results indicate that guanosine could attenuate A_2A_R signaling (i.e., agonist-mediated cAMP accumulation and cell impedance responses) in an A_1_R-dependent manner (Figure 6B). On the other hand, when the A_1_R-dependent signaling (i.e., intracellular Ca^2+^ mobilization) was assessed, guanosine was unable to modulate receptor’s function both in singly and doubly A_1_R-A_2A_R transfected cells. Taken together, our results suggest that while guanosine did not signal through A_1_R, it requires this receptor to exert its A_2A_R modulatory effect, which could indicate that the A_1_R/A_2A_R heteromer might be a molecular substrate for guanosine.

The A_1_R/A_2A_R heteromer displays some functional characteristics similar to that reported for other AR-containing oligomers, for instance A_2A_R combined with the dopamine D_2_ receptor (D_2_R) or the cannabinoid CB_1_ receptor (CB_1_R) [37]. Interestingly, these receptor heteromers have been shown to exert reciprocal receptor-receptor allosteric antagonistic interactions [38]. Precisely, an A_1_R/A_2A_R heteromer-mediated transmembrane-dependent negative allosteric interaction at the ligand-receptor binding level has been described [39]. In addition, co-activation of both receptors led to a canonical protein Gs-Gi antagonistic interaction at the level of the adenylyl cyclase [40]. This situation makes it difficult to conclude whether an effect in a given signaling pathway is caused by either the allosteric or the canonical interaction. Thus, our data showing that guanosine was able to modulate AR functioning (i.e., cAMP assay) only in cells expressing A_1_R and A_2A_R do not permit a clear determination of the interaction at the intracellular level (i.e., canonical protein Gs-Gi antagonistic interaction). However, considering the whole picture, it seems likely that guanosine effects in the physiological context may depend on the co-expression of both receptors and their and interaction. Indeed, guanosine did not disrupt the A_1_R/A_2A_R heteromer, as observed by a saturable BRET signal, similar to that obtained following adenosine treatment, and by membrane co-localization of A_1_R and A_2A_R in guanosine-treated cells (Figure A1).

Overall, our data suggest an important role for the A_1_-A_2A_ receptor–receptor interaction in guanosine-mediated effects. Thus, while our results seem to rule out an eventual guanosine-mediated A_1_R-A_2A_R canonical antagonistic interaction, further investigation is needed to ascertain whether guanosine may either modulate the well-known A_1_R-A_2A_R allosteric interaction or an indirect mechanism of action yet to be discovered.

## 5. Conclusions

In summary, our results revealed that certain AR-related guanosine-mediated effects rely on A_1_R and A_2A_R co-expression. Indeed, in ex vivo experiments, the well-known guanosine-mediated neuroprotective effect depends on A_2A_R expression. Thus, guanosine failed to protect A_2A_R^−/−^ mouse hippocampal slices from ischemia-induced damage. In addition, while guanosine did not interfere with A_1_R-mediated signaling, it modulated A_2A_R binding and intracellular signaling only in A_1_R-A_2A_R co-expressing cells. Overall, our results suggest that A_1_R and A_2A_R may constitute a molecular substrate involved in guanosine effects, but the precise mechanism of action of guanosine involving ARs still is intriguing.

## Figures and Tables

**Figure 1 cells-08-01630-f001:**
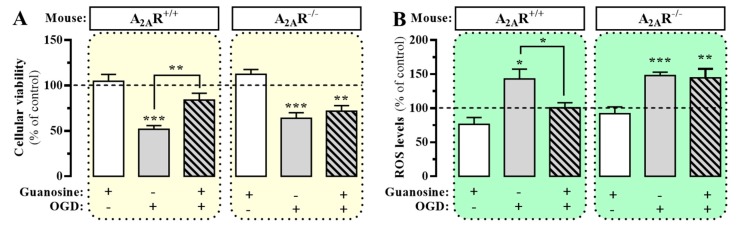
Guanosine-mediated neuroprotection in mouse hippocampal slices. Hippocampal slices from A_2A_R^+/+^ and A_2A_R^−/−^ mice were subjected to oxygen/glucose deprivation (OGD) in the absence or presence of guanosine (100 µM) for 15 min before, and during OGD and re-oxygenation. The cellular viability (**A**) was assessed by MTT reduction whereas ROS levels (**B**) were measured after incorporation of the DCFDA fluorescent probe. Results were normalized to the control slices (vehicle-treated slices, dashed line) and expressed as mean ± SEM of three independent experiments performed in triplicate. The asterisks indicate statistically significant differences (* *p* <0.05, ** *p* < 0.01 and *** *p* < 0.001; one-way ANOVA with Tukey’s post-hoc test).

**Figure 2 cells-08-01630-f002:**
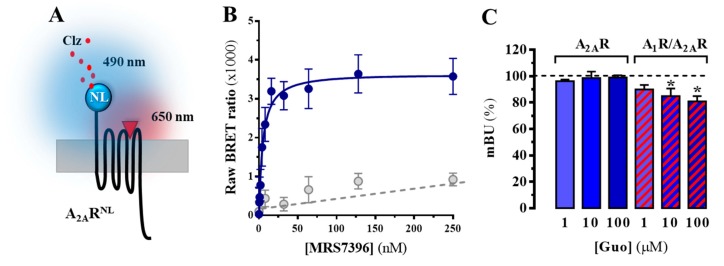
NanoBRET-based A_2A_R binding determinations. (**A**) Schematic representation of the NanoBRET-based assay using A_2A_R^NL^ stably expressing cells and the fluorescent MRS7396 ligand (red triangle). When the coelenterazine (Clz) substrate is metabolized by NanoLuciferase (NL), its 475 nm light emission may engage in a BRET process with MRS7396 given the close proximity (i.e., bound to A_2A_R^NL^). (**B**) NanoBRET signal for A_2A_R^NL^ with increasing MRS7396 concentrations in the absence (solid line) and presence (dotted line) of 1 μM SCH442416. (**C**) Guanosine (Guo) effects on MRS7396 binding to cells expressing A_2A_R^NL^ (blue bars) or A_2A_R^NL^ plus A_1_R^SNAP^ (red dashed bars). Cells were incubated with MRS7396 (10 nM) and increasing guanosine concentrations (1–100 μM) in the presence or absence of 1 μMSCH442416 to allow specific binding calculations. Results were normalized to the MRS7396 specific binding in the absence of guanosine for each transfection set and expressed as mean ± SEM of four independent experiments performed in triplicate. The asterisks indicate statistically significant differences * *p* < 0.05, one-way ANOVA followed by Dunnett’s post-hoc testwhile compared to control (dashed line).

**Figure 3 cells-08-01630-f003:**
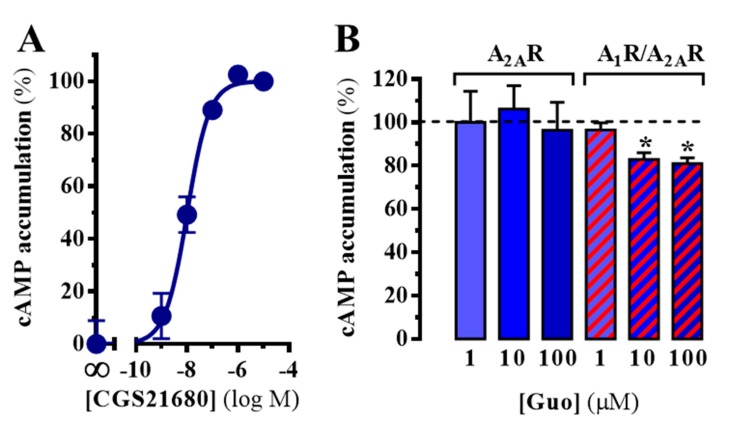
A_2A_R-dependent cAMP accumulation. (**A**) Concentration-dependent effects of CGS21680 in cAMP accumulation in singly A_2A_R expressing cells. The signal was normalized by assigning the 100% to the maximum signal obtained and 0% to cells without ligand. The data are expressed as the mean ± SD of a representative experiment performed in triplicate. (**B**) Guanosine effects on CGS21680-mediated cAMP accumulation in cells expressing A_2A_R^SNAP^ (blue bars) or A_2A_R^SNAP^ plus A_1_R^SNAP^ (red dashed bars). Results were normalized to the specific cAMP accumulation in the absence of guanosine for each transfection set and are expressed as mean ± SEM of four independent experiments performed in triplicate. The asterisks indicate statistically significant differences * *p* < 0.05, one-way ANOVA followed by Dunnett’s post-hoc testwhile compared to control (dashed line).

**Figure 4 cells-08-01630-f004:**
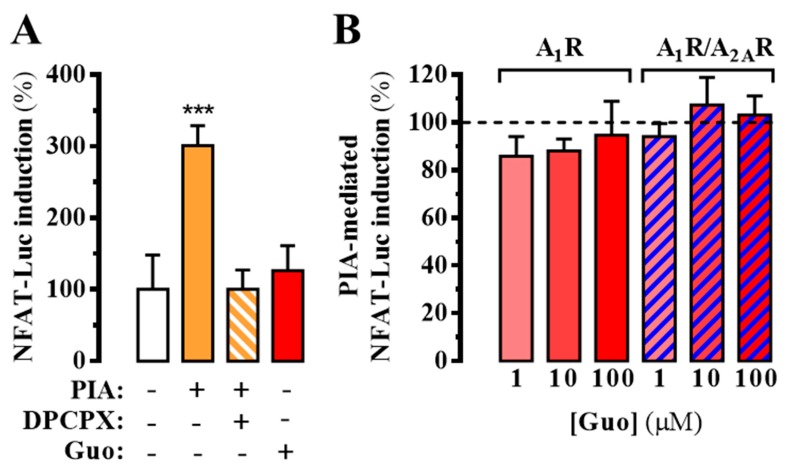
A_1_R-dependent intracellular Ca^2+^mobilization. (**A**) Determination of A_1_R-mediated intracellular calcium accumulation by means of a luciferase reporter assay system. HEK-293T cells were transiently transfected with the firefly luciferase-encoding plasmid (pGL4-NFAT-luc2p) and the cDNAs encoding the A_1_R^SNAP^ and the YFP. Thirty-six hours after transfection, cells were treated 6 h with the A_1_R agonist R-PIA (PIA, 50 nM) in the absence or presence of DPCPX (500 nM) or guanosine (Guo, 100 μM). Light emission is presented as the percentage increase over basal levels. The data are expressed as the mean ± SEM of three independent experiments performed in triplicate. The asterisks indicate statistically significant differences *** *p* < 0.001, one-way ANOVA followed by Dunnett’s post-hoc test when compared to control. (**B**) Guanosine modulation of R-PIA-mediated intracellular Ca^2+^mobilization (PIA-mediated NFAT-Luc induction) in cells expressing A_1_R^SNAP^ (red bars) or A_1_R^SNAP^ plus A_2A_R^SNAP^ (blue dashed bars). The dotted line represents the Ca^2+^ mobilization induced by R-PIA in the absence of guanosine within each cell transfection group. The data are expressed as the mean ± SEM of three independent experiments performed in triplicate.

**Figure 5 cells-08-01630-f005:**
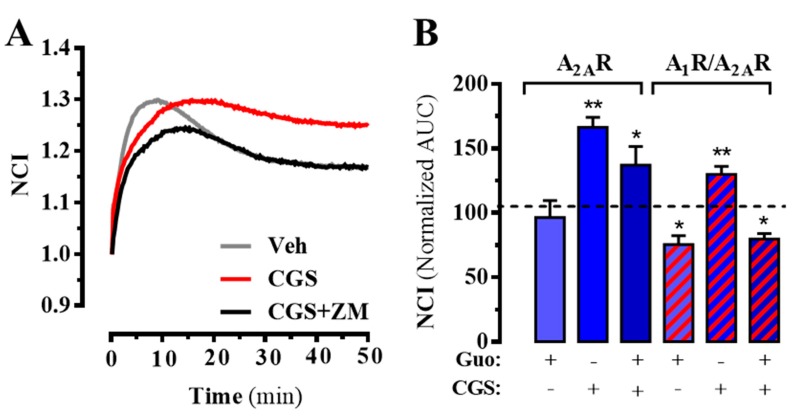
A_2A_R-mediated whole-cell label-free responses. (**A**) Real-time cellular impedance changes upon CGS21680 (200 nM) incubation in the absence or presence of ZM241385 (1 µM). The signal was normalized when the ligand was added. (**B**) Guanosine (100 µM) effects on CGS21680-mediated cellular impedance changes in cells expressing A_2A_R^SNAP^ (blue bars) or A_2A_R^SNAP^ plus A_1_R^SNAP^ (dashed red bars). Results are presented as area under the cure (AUC) and normalized to the AUC in the basal condition (i.e., absence of any drug) for each transfection set and expressed as mean ± SEM of three independent experiments performed in duplicate. * *p* < 0.05 and ** *p* < 0.01, one-way ANOVA followed by Dunnett’s post-hoc test while compared to control (dashed line).

**Figure 6 cells-08-01630-f006:**
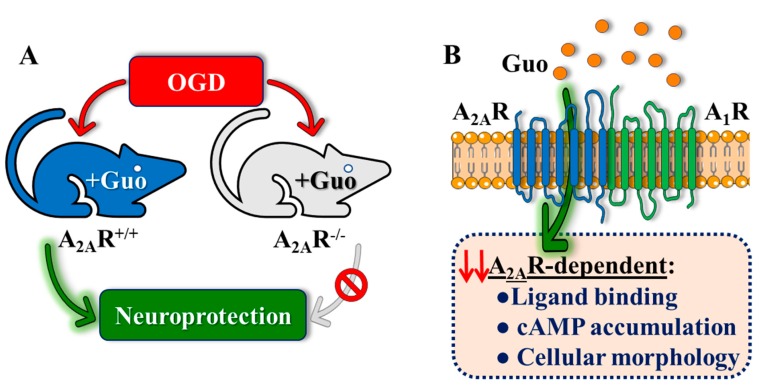
Schematic summary of the overall findings. (**A**) Guanosine-mediated neuroprotection in mouse is dependent on A_2A_R expression. Thus, guanosine fails to neuroprotect from OGD damage in A_2A_R^−/−^ mouse hippocampal slices. (**B**) Guanosine modulates A_2A_R functionality in living cells in an A_1_R-dependent manner. While guanosine does not interfere with A_1_R-dependent signaling, it modulates A_2A_R binding and intracellular signaling (i.e., cAMP accumulation and cellular morphology) only in A_1_R-A_2A_R co-expressing cells. Therefore, A_1_R and A_2A_R may constitute a molecular substrate involved in guanosine-mediated effects, but the precise mechanism of action of guanosine involving ARs is still lacking.

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
