# Peer review of "Adenosine A1-A2A Receptor-Receptor Interaction: Contribution to Guanosine-Mediated Effects"

_cells, 2019, doi:10.3390/cells8121630_

Round 1

Reviewer 1 Report

Authors investigated neuroprotective effects of nucleoside guanosine in ichemia animal and cellular model. They found that guanosine prevened the reduction of cellular viability and increase in free radical generation by OGD in wild-type but not in A2A receptor KO mice. Guanosine did not modify MRS7396 binding to A2A  receptor, but partially blocked it in cells co-expressing A1 and A2A receptors. In functional assays, guanosine blocked A2A receptor mediated cAMP accumulation and calcium mobilization only in cells co-expressing A1 and A2A receptors. It also reduced A2A agonist mediated cell impedance responses in these cells. Authors conclude that guanosine effects need A1-A2A receptor complexes in presented models.

The paper is very clearly written and experiments are carefully planned. Data are well described and discussed. Techinques used are modern. Overall, the data are important to explain the neuroprotective mechanism of guanosine.

Author Response

Authors investigated neuroprotective effects of nucleoside guanosine in ischemia animal and cellular model. They found that guanosine prevented the reduction of cellular viability and increase in free radical generation by OGD in wild-type but not in A2A receptor KO mice. Guanosine did not modify MRS7396 binding to A2A receptor, but partially blocked it in cells co-expressing A1 and A2A receptors. In functional assays, guanosine blocked A2A receptor mediated cAMP accumulation and calcium mobilization only in cells co-expressing A1 and A2A receptors. It also reduced A2A agonist mediated cell impedance responses in these cells. Authors conclude that guanosine effects need A1-A2A receptor complexes in presented models. The paper is very clearly written, and experiments are carefully planned. Data are well described and discussed. Techniques used are modern. Overall, the data are important to explain the neuroprotective mechanism of guanosine.

Answer: We really appreciate the positive comments of the referee.

Reviewer 2 Report

General comments

 Manuscript employs novel methodologies. Overall, language editing seems necessary. Several lines show the need for grammatical enhancement and there are many phrases or words that may confuse readers.   Since this paper attempts to unveil a novel role for adenosine receptor A1 and A2a in mediating guanosine effects, it would be beneficial to include a summary schematic to help readers understand the overall findings. This study had access to A2A knockout mice, which were not used to the full capacity. Cells could have been extracted or isolated from the mice for further studies (ex-vivo analysis).

Detailed comments

Page 2, Line 55-58 – This sentence should be presented in a more concise and grammatically accurate manner.

Page 2, Line 59-62 – These statements need to be re-arranged or edited to make them more cohesive.

Page 5, Line 211-213 – How stable is guanosine in the buffer? How did you determine the concentration of guanosine?

Page 5, Line 210, 212, 215 – Figure 1C and 1D are not present.

Page 6, Line 250-252 and Page 7 Line 274-276 – Significant reduction was noted in the figures, but these are described as “partial effect.”

Figure 2 – Why is there such a drastic difference in concentrations of MRS7396 tested in Figure 2B (0-250 uM) vs. 2C (10 nM)?

Figure 3A – Error bars for the last three data points are not shown.

Figure 4A – How about the effect of adding PIA and Guo together? Will it show any synergistic effect?

Figure 5A – How significant is the increase in real-time cellular impedance changes with CGS21680 incubation?

Figure 5B – The way authors normalized this particular result was not consistent with the other ones in the paper (Figure 1-4). In other words, other figures showed 100% being the control, however the control for Figure 5B is hard to discern. Please clarify.

Author Response

General comments

 Manuscript employs novel methodologies. Overall, language editing seems necessary. Several lines show the need for grammatical enhancement and there are many phrases or words that may confuse readers. Since this paper attempts to unveil a novel role for adenosine receptor A1 and A2A in mediating guanosine effects, it would be beneficial to include a summary schematic to help readers understand the overall findings. This study had access to A2A knockout mice, which were not used to the full capacity. Cells could have been extracted or isolated from the mice for further studies (ex-vivo analysis).

Answer:

We thank to the referee for the constructive comments on our manuscript. As requested, the grammar of the entire manuscript has been revised, thus an English native speaker has revised the text. In addition, we generated a scheme summarizing the overall findings of our work (see Figure 6). Finally, we agree with the referee that the use of the A2AR-/- mouse has been limited to certain experiments. Indeed, we decided to use hippocampal slices subjected to oxygen/glucose deprivation due to our previous information regarding the neuroprotective effect of guanosine in this preparation. The A1R- and A2AR-dependent pharmacological neuroprotection in hippocampal slices was shown previously, (Dal-Cim et al., J. Neurochem. Journal of Neurochemistry 2013, 126, 437–50). Thus, here we aimed to unequivocally demonstrate the specific contribution of A2AR using hippocampal slices from an A2AR deficient mice which were subjected to oxygen/glucose deprivation (OGD). Indeed, the referee is right, we could have used primary hippocampal neurons from A2AR-/- submitted to OGD to demonstrate the A2AR-dependent neuroprotection but we used a simpler cellular model as transfected HEK293 cells.

Detailed comments

Page 2, Line 55-58 – This sentence should be presented in a more concise and grammatically accurate manner.

Answer:

We amended this statement accordingly (“Brain ischemia is one of the major health disability conditions worldwide [3]. It occurs after blood supply collapse that leads to a reduced level of oxygen and glucose within the affected brain area. Similarly, upon excitotoxicity and oxidative stress a failure of cellular bioenergetics occurs [4]. Importantly, in animal and cellular models of ischemia, excitotoxicity and oxidative stress a neuroprotective role of guanosine has been extensively reported [5–10]”).

Page 2, Line 59-62 – These statements need to be re-arranged or edited to make them more cohesive.

Answer:

We amended this statement accordingly (“Indeed, we have demonstrated that guanosine prevents reactive oxygen species (ROS) generation and cell death in hippocampal slices subjected to the oxygen/glucose deprivation (OGD) [11]”).

Page 5, Line 211-213 – How stable is guanosine in the buffer? How did you determine the concentration of guanosine?

Answer:

The referee highlighted an important question here. The Krebs-Ringer buffer, largely used in ex vivo experiments, was utilized to maintain the hippocampal slices at physiological osmolarity and pH, thus mimicking an artificial cerebrospinal fluid. Guanosine is a rather stable molecule in aqueous solution (Meth. Enz. Anal. 1974, 4, 1928-1931). Indeed, we used a concentration of 100 µM of guanosine as we already determined its physiological effect in similar ex vivo experiments (Oleskovicz et al, Neurochem Int, 52, 411-418; 2008; Dal-Cim et al., Journal of Neurochemistry 2013, 126, 437–50; Thomaz et al, Purinergic573 Signalling2016, 12, 707–718).

Page 5, Line 210, 212, 215 – Figure 1C and 1D are not present.

Answer:

We thank to the referee for noticing this mistake. Thus, we amended accordingly.

Page 6, Line 250-252 and Page 7 Line 274-276 – Significant reduction was noted in the figures, but these are described as “partial effect.”

Answer:

The referee is right, we should better describe the data shown in Figure 2C. Thus, we adapt the text to: “Notably, in A1R-A2AR doubly-expressing cells, guanosine (100 µM) was able to significantly reduce a 19 ± 4% (P = 0.0138) the binding of MRS7396 to the A2ARNL, thus indicating that the A1R/A2AR heteromer might play a potential role in AR-related guanosine-dependent effects (Figure 2C)”. In addition, we changed the description of Figure 3B (“Conversely, in cells doubly expressing A1R and A2AR, guanosine guanosine (100 µM) was able to significantly reduce a 19 ± 3% (P = 0.0460) the A2AR-mediated cAMP accumulation (Figure 3B)”)

Figure 2 – Why is there such a drastic difference in concentrations of MRS7396 tested in Figure 2B (0-250 uM) vs. 2C (10 nM)?

Answer:

We apologize for this typing error. In the graphic shown in Figure 2B the units for MRS7396 concentration (i.e. X axis) should be nM.

Figure 3A – Error bars for the last three data points are not shown.

Answer:

The referee is right, the error bars of the last three data points are not visible. The reason is that the standard deviation (SD) for these three points are too small (i.e. -7: 89 ± 1%; -6: 103 ± 2% and -5: 100 ± 1%).

Figure 4A – How about the effect of adding PIA and Guo together? Will it show any synergistic effect?

Answer:

We apologize for the misunderstanding, but the condition of R-PIA plus Guo is shown in Figure 4B. Thus, in this figure the dotted line represents the Ca2+ mobilization induced by R-PIA in the absence of guanosine within each cell transfection group and taken as 100%. Now, we definedn the Y axis as “PIA-mediated NFAT-Luc induction” to avoid any confusion.

Figure 5A – How significant is the increase in real-time cellular impedance changes with CGS21680 incubation?

Answer:

In the Figure 5A we show a representative real-time impedance trace for vehicle, CGS and CGS+ZM treated cells. The significant impedance increase induced by CGS is shown in Figure 5B and commented in the results section as: “Interestingly, addition of CGS21680 resulted in a significant (P = 0.015) increase of impedance, which was blocked by incubation with the selective A2AR antagonist ZM241385 (Figure 5A and B).”

Figure 5B – The way authors normalized this particular result was not consistent with the other ones in the paper (Figure 1-4). In other words, other figures showed 100% being the control, however the control for Figure 5B is hard to discern. Please clarify.

Answer:

The referee is right. With this graphic representation we intended to show more clearly the differences between the different experimental groups. However, since this may generate confusion, we already change the representation of the data shown in Figure 5B to agree with the normalization (100%) shown in Figures 1-4. We apologize for the inconvenience.

Reviewer 3 Report

The paper is about a very hot topic regarding the function of adenosine A1 and A2A receptor interaction. The paper intends to demonstrate the mechanisms involved into the protective effect exerted by guanosine during glucose and oxygen deprivation in hippocampal slices. A number of even sophisticate technical approaches were utilized to meet this goal, however I believe that some very important cellular mechanisms were not taken into account. Guanosine is a purine nucleoside, and, similarly to other nucleosides can enter the cell through different transport systems belonging to two different families: concentrative and equilibrative. Once the compound is inside the cell it is phosphorolytically cleaved  into guanine (a purine base) and ribose-1-phosphate, a sugar that can be readily oxidized (see Uptake and utilization of nucleosides for energy repletion. Giannecchini M, Matteucci M, Pesi R, Sgarrella F, Tozzi MG, Camici M. Int J Biochem Cell Biol. 2005 Apr;37(4):797-808.). Since the paper is about a protective effect exerted by guanosine against glucose deprivation, the possibility that guanosine is exerting this effect through its intracellular catabolism must be taken in account or, at least, discussed. There are several questions without answer such as:

is the protective effect specifically exerted by guanosine?, can inosine obtain the same result? (also inosine can serve as ribose-1-phosphate donor  in energy repletion)

are you sure that the compounds utilized as agonist or antagonist of adenosine receptors are not inhibitors of nucleoside transport systems or of nucleoside catabolic enzymes ? (in some cases it has been demonstrated that  agonists and antagonists of adenosine receptors are  inhibitors of equilibrative transport systems ( see Interaction of nucleoside analogues with nucleoside transporters in rat brain endothelial cells.Chishty M et al. J Drug Target. (2004)).

The effect exerted by guanosine is  affected by inhibitors of nucleoside transport systems?

Author Response

The paper is about a very hot topic regarding the function of adenosine A1 and A2A receptor interaction. The paper intends to demonstrate the mechanisms involved into the protective effect exerted by guanosine during glucose and oxygen deprivation in hippocampal slices. A number of even sophisticate technical approaches were utilized to meet this goal; however, I believe that some very important cellular mechanisms were not taken into account. Guanosine is a purine nucleoside, and, similarly to other nucleosides can enter the cell through different transport systems belonging to two different families: concentrative and equilibrative. Once the compound is inside the cell it is phosphorolytically cleaved into guanine (a purine base) and ribose-1-phosphate, a sugar that can be readily oxidized (see Uptake and utilization of nucleosides for energy repletion. Giannecchini M, Matteucci M, Pesi R, Sgarrella F, Tozzi MG, Camici M. Int J Biochem Cell Biol. 2005 Apr;37(4):797-808.). Since the paper is about a protective effect exerted by guanosine against glucose deprivation, the possibility that guanosine is exerting this effect through its intracellular catabolism must be taken in account or, at least, discussed.

Answer:

We thank to the referee for the positive evaluation of the manuscript. Indeed, we understand the referee’s concern regarding potential cellular mechanisms behind the guanosine neuroprotective effects against OGD. We have previously demonstrated that guanosine-mediated neuroprotective and neurotrophic effects were somehow extracellular. Certainly, guanosine can be metabolized into guanine and ribose-1-phosphate. However, we previously demonstrated that the neuroprotective effect of guanosine against OGD-induced damage still is observed even in the presence of nucleoside transporters inhibitors (Olescovicz et al., 2008), thus suggesting that this guanosine effect depends on an extracellular site of action. Accordingly, here we aimed to assess the putative involvement of A2AR and A1R in this guanosine-mediated neuroprotection. Finally, the Giannecchini et al. manuscript indicated that inosine effects of ribose replenishment are observed at millimolar concentrations of (deoxy)inosine or inosine, thus suggesting a metabolic fuel effect rather than a modulatory role.

There are several questions without answer such as:

-is the protective effect specifically exerted by guanosine?, can inosine obtain the same result? (also inosine can serve as ribose-1-phosphate donor  in energy repletion)

Answer:

The referee highlighted a very important question here. As early stated, here we aimed to assess the role of adenosine in guanosine-mediated effects. Therefore, we didn’t evaluate the neuroprotective effects of inosine and whether the adenosine receptors are involved with. Interestingly, as the referee suggested, it can’t be rule out that inosine may have a protective effect against OGD-induced damage. Indeed, Jurkowitz et al. (J. Neurochem. 1998, 71:535) demonstrated that inosine had a protective although at higher concentrations to that of guanosine used here. Interestingly, while inosine was suggested to be a neuromodulator, its effects were limited to be A1R-dependent and not related to A2AR expression (Nascimento et al. 2015, Mol.Neurobiol. 51:1368–1378). Overall, inosine may be reminisce guanosine neuroprotection since some effects are adenosine receptors-dependent as well.

-are you sure that the compounds utilized as agonist or antagonist of adenosine receptors are not inhibitors of nucleoside transport systems or of nucleoside catabolic enzymes? (in some cases it has been demonstrated that agonists and antagonists of adenosine receptors are inhibitors of equilibrative transport systems (see Interaction of nucleoside analogues with nucleoside transporters in rat brain endothelial cells.Chishty M et al. J Drug Target. (2004)).

Answer:

Again, the referee pointed out an important question. Indeed, the ability of adenosine receptors ligands to modulate adenosine transport has been largely explored. For instance, some A2AR agonists, including CGS21680, were examined for their ability to activate A2AR and inhibit nucleoside transport (Balwierczak et al. 1989, J. Pharmacol. Exp. Ther. .251:279-87). In this paper the authors described that 20 μM CGS21680 was unable to preclude [3H]uridine transport into guinea pig erythrocytes. In our study we used CGS21680 at 200 nM in HEK293 cells expressing both A1R and A2AR. Thus, it seems reasonable that our CGS21680-mediated effects are not due to nucleoside transport blocking. In addition, these authors report that R-PIA had a transport inhibition EC50 of 3 μM. Again, the concentration of R-PIA used in HEK293 cells expressing both A1R and A2AR was of 50 nM, far away to affect nucleoside transport. Interestingly, Chishty M et al. (J Drug Target. 2004) reported that some adenosine receptor ligands (i.e. GR 150185, GR 395849, GR 56072X, GR 56071X, CCI 4019, GR 66683X, GR 242468, GR 79236X) showed an adenosine transport inhibition ranging from 10 μM to 100 μM, thus far from the concentrations we used in our cellular model. Overall, we believe that adenosine receptor ligand would not alter nucleoside transport.

-The effect exerted by guanosine is affected by inhibitors of nucleoside transport systems?

Answer:

This question might be related to the previous one. Thus, as abovementioned, the incubation with a non-selective broad-spectrum (CNTs and ENTs) nucleoside inhibitor dipyridamole, does not affect the protective effect of guanosine in the same OGD protocol using hippocampal slices (Oleskovicz et al. Neurochem. Int. 2008, 52, 411-418;).

In addition, dipyridamole did not affect guanosine-mediated effects in other experimental paradigms: i) Guo-mediated trophic effects in primary cultured cerebellar neurons (Decker et al. 2019, Purinergic Signal); ii) Guo-mediated reorganization of extracellular matrix proteins (favouring neuronal adhesion) in cerebellar astrocytes in culture (Decker et al. 2007, J Neurosci Res, 85, 1943-51). Also, dipyridamole and adenosine deaminase preincubation did not alter the cytotoxic effect of guanosine on glioma cells (Oliveira et al. 2017, Purinergic Signal , 13, 305-318). Overall, these evidences suggested  an extracellular action of guanosine-evoked neuroprotective and neurotrophic effects involving A2AR expression.

Round 2

Reviewer 3 Report

The authors answered to all my observations in a convincing way. I still believe that the possibility of a contribution in neuroprotection exerted by intracellular catabolism of guanosine cannot be ruled out and that this deserve at least a mention

discussing the data. In my experience when an animal model silenced for a particular protein is developed, the result is always a model with several genes expression alterations, to compensate for the loss of that particular protein, so your model is never completely under control. Anyway, I agree for the publication of this manuscript in its present form.